# The Productive, Economic, and Social Efficiency of Vineyards Using Combined Drought-Tolerant Rootstocks and Efficient Low Water Volume Deficit Irrigation Techniques under Mediterranean Semiarid Conditions

**Pascual Romero Azorín * and José García García**

Irrigation and Stress Physiology Group, Department of Bio-Economy, Water and Environment, Instituto Murciano de Investigación y Desarrollo Agrario y Alimentario (IMIDA), c/Mayor s/n, La Alberca, 30150 Murcia, Spain; jose.garcia21@carm.es
* Correspondence: pascual.romero@carm.es

**Abstract:** In many areas of southern Europe, the scarcity of water due to climate change will increase, making its availability for irrigation an even more limiting factor for agriculture. One of the main necessary measures of adaptation of the vineyards in these areas will be the implementation of water-saving irrigation strategies and technologies to improve WUE (water use efficiency). The objective of the present study was to evaluate the long-term economic viability/profitability of different deficit irrigation techniques such as regulated deficit irrigation (RDI) and partial root-zone irrigation (PRI) with low water volume/fertilizer applied in a Monastrell vineyard in southeastern Spain to plants grafted on different rootstocks, and to assess the productive, social, and economic efficiency in these semiarid conditions. Through a cost/benefit analysis, socio-economic and environmental criteria for the selection of optimal deficit irrigation strategies and tolerant/water use efficient rootstocks for the vineyards in arid environments are proposed. Our analysis shows a clear conflict between productivity and quality in wine grape production. Productive and economic indices, such as yield, productive WUE (kg m$^{-3}$), economic efficiency (€ m$^{-3}$), break-even point (kg ha$^{-1}$), and water productivity (€ m$^{-3}$), were inversely related with berry quality. Besides, high berry quality was closely related with higher production costs. Under the current market of low-priced grapes, if the grower is not rewarded for the quality of the grapes (considering technological, phenolic, and nutraceutical quality), the productivity vision will continue and the cost-effective option will be to produce a lot of grapes, even if at the expense of the berry and wine quality. In this situation, it will be difficult to implement optimized deficit irrigation strategies and sustainable irrigation water use, and the pressure on water resources will increase in semiarid areas. Public policies should encourage vine growers to invest in producing high-quality grapes as a differentiating character, as well as to develop agronomic practices that are environmentally and socially sustainable, by the grapes more adjusted to their real quality and production costs. Only in this way we can implement agronomic measures such as optimized low-input DI (deficit irrigation) techniques and the use of efficient rootstocks to improve WUE and grape quality in semiarid regions in a context of climate change and water-limiting conditions.

**Keywords:** cost/benefit; partial root–zone irrigation (PRI); regulated deficit irrigation (RDI); profitability; water economic efficiency; water use efficiency; water social efficiency

## 1. Introduction

Recent studies project greater warming and more severe water shortage in the south of Europe, especially in the Iberian Peninsula, and, more particularly, the south and southeast of the Iberian Peninsula, as a result of climate change [1–3]. In addition, almost all simulations for the Mediterranean Basin foresee that warming will exceed the average for global warming. Global warming above 2 °C (with respect to the pre-industrial era) may involve very important changes in Mediterranean ecosystems, such as a loss of biodiversity, reduction of forest areas, and the expansion of desert areas (increased desertification in southeastern Spain), as well as bringing major risks to the population as a result of scarcity of water resources and an increase in the demand for water for irrigation, energy, and domestic use [4]. As regards wine production, according to climatic projections for Europe and Spain [3,5], the southern regions of Europe and the Mediterranean arc, especially the South and East of the Iberian Peninsula, will need the most effort in order to adapt, with increased costs to maintain the quality and productivity of vineyards, since these regions will face changes of greater magnitude than other wine-producing areas [6]. For example, a study conducted to explore the possible measures of adaptation to climate change in several Spanish wine-producing regions points to the fact that the Protected Designation of Origin (PDO) of Jumilla and La Mancha are two of the most vulnerable, and that they may suffer a high impact due to a great increase in the projected temperature and a decrease in precipitation [6]. In these areas, scarcity of water will increase, making the availability of irrigation water an even more limiting factor for agriculture. The increase in temperature will generate a water shortage at the atmospheric level, which will produce an increase in the rate of evapotranspiration of around 75–125 mm by the 2050s for most of Europe [7]. In this scenario, there will be an increase in water needs of vines, since irrigation will be necessary to maintain a vineyard's long-term sustainability and to prevent severe stress in many wine-producing regions in the south of the peninsula [8]. The three main measures of adaptation to climate change that will have to be taken in highly vulnerable regions are the selection of varieties and rootstocks that are more tolerant to drought and high temperatures, changes in soil management practices, and increased irrigation [6]. Although many Mediterranean vineyards are currently cultivated on dry land, one of the main measures of the adaptation of the vineyards in these areas will necessarily be the implementation of an efficient irrigation system, with important changes in water management through the implementation of water-saving irrigation strategies, techniques, and technologies to improve the efficiency in the use and application of irrigation water. Two of the most promising deficit irrigation (DI) techniques in vineyards with the greatest potential in semiarid regions to increase water use efficiency and improve the quality of the berry and wine are Regulated Deficit Irrigation (RDI) and Partial Root Drying Irrigation (PRI) [9–13]. Also, the use of rootstocks with different degrees of vigor and sensitivity to water deficit may be considered an important and useful agronomic tool for the efficient management of the vineyard when applying RDI and PRI, selecting rootstocks that are better adapted to the application of both in order to optimize these irrigation techniques in semiarid conditions [14,15].

In the Mediterranean Basin, viticulture plays a vital role in the socio-economic life of the region, often lacking other viable economic alternatives. In the southeast of Spain, together with almonds, woody crops have the greatest importance as an agroforestry contribution, and any reduction may lead to abandonment and the consequential problems of erosion and desertification [16,17]. The situation of vineyards is particularly serious in regions with very limiting climatic conditions, such as the shortage of rainfall. A productive specialization according to the destination of the grape to QWpsr (Quality Wines Produced in Specified Regions) and a consequent differentiation of the product depending on the quality and their environmental and landscape features could make wine-producing a viable activity linked to the rural environment [16,18].

Several studies have assessed the efficiency in the use of water from a productive standpoint [19,20], but there are few works that have evaluated this efficiency from a social or economic perspective [21–23], with the importance being that using different indexes of socio-economic efficiency involves the need for economic studies that could serve as a support for decision making. The economic analysis of water

resources illustrates the need for a global perspective of economic efficiency, i.e., not only technical or productive efficiency [23]. In summary, it seems essential to identify the conditions in which irrigation strategies may be economically justified in the long term.

The assessment of economic sustainability is obviously a prerequisite to carry out business operations, but an assessment of the sustainability of the environment may also be a strategic tool that can help increase the value of the product. In the last two decades, worldwide awareness of the importance of the environment has grown dramatically [24]. Consumers have included environmental concerns as an important factor in their purchasing processes, selecting those products that show sensitivity towards the environment [24], and distribution chains have responded promptly to this consumer demand. The establishment of sustainable production patterns based on socio-economic and environmental criteria is a key strategy toward viable and competitive wine production. It is necessary to establish cultivation systems and production in the winery that make cultivation sustainable by promoting the quality of the wine grape and by implementing working methods with favorable effects on the social, economic, and environmental levels for rural populations and environments.

The current situation, whereby the method of payment in many areas is still kg/°Baumé, without taking into account other quality parameters, favors high productivity at the expense of quality. Improvements in grape quality are not taken into account and, in most cases, are not reflected in the higher prices of the grapes, so there is little financial reward for growers who offer quality [18]. Many studies have shown that increases in water supplied through irrigation increase production [18,25], and if, in addition, growers are paid on the basis of production and not quality, a productivity view prevails. In many cases, this has favored increased irrigation and the application of full irrigation strategies to obtain high productivity at the expense of grape quality [18].

The Protected Designation of Origin (PDO) constitutes the system used in Europe to differentiate quality, both in vines and wine. In general, most PDOs establish limitations on productivity (production ceilings). In southeastern Spain, for example, the PDO production limitation is around 7000–9000 kg/ha for red grapes (PDO Jumilla, PDO Bullas, PDO Yecla, PDO Alicante, PDO Valencia, etc.). When yields exceed the authorized total, the production may not be marketed under those names and thus fall into the category of table wines (of lower quality), the lowest level recognized by law for vines and wines. However, there is usually a significant improvement of grape and wine quality when RDI (Regulated Deficit Irrigation) or PRI (Partial Root Drying Irrigation) is applied, mainly because of an increase in the content of polyphenols and nutraceuticals in berries and wines [12–15,26,27]. At present, maturity control indexes (sugar and acids in grapes) are clearly insufficient to evaluate the quality of grapes [28]. Thus, anthocyanins and other polyphenolic compounds and nutraceuticals related to the color and flavor, and other healthy aspects play an important role in the quality of the grapes and wines. This translates into an improvement in the organoleptic characteristics of wine, such as color, aroma, and flavor, which is of great commercial and economic importance. In arid areas with very restrictive conditions (low water availability and high price of irrigation water), the commitment to higher quality associated with dry land cultivation or with RDI strategies and a consequent payment at which differentiated quality would make the viticulture viable and profitable [16,17].

The objective of the present study was to evaluate the long-term socioeconomic viability/ profitability of different DI (deficit irrigation) techniques (i.e., RDI and PRI) with low water volumes applied in a vineyard of Monastrell in southeastern Spain, grafted on different rootstocks, and to assess the productive, social, and economic efficiency in these semiarid conditions. Through a cost/benefit analysis, socio-economic, quality, and environmental criteria for the selection of optimal deficit irrigation strategies and drought-tolerant and water use-efficient rootstocks for the vineyards in arid and semiarid environments are proposed. Based on berry and wine quality criteria, optimum ranges of yield and WUE are proposed under current grape market conditions to look for a compromise between productivity, quality, and returns for the grower.

## 2. Material and Methods

### 2.1. Experimental Conditions, Plant Material, and Irrigation Treatments

This research was carried out from 2012 to 2017 (six years) in a 0.4 ha vineyard at the Instituto Murciano de Investigación y Desarrollo Agrario y Alimentario (IMIDA) experimental station in Cehegín, Murcia, southeastern Spain (38° 6´ 38.13" N, 1° 40´ 50.41" W, 432 m above sea level). The soil was an 80-cm-deep clay loam (33% clay, 38% silt, and 30% sand) with 1.12% of organic matter. The climate is Mediterranean semiarid, with long hot and dry summers and scarce annual rainfall (around 386 mm·year$^{-1}$), with reference evapotranspiration (ETo) above 1200 mm [14]. The grapevines (*Vitis vinifera* L, var. Monastrell, syn. Mourvedre, a local red wine variety) were 20+ years old and were grafted on five different commercial rootstocks, each with a different vigor and drought tolerance: 140Ru (*V. rupestris* x *V. berlandieri*), 1103P (*V. rupestris* x *V. berlandieri*), 41B (*V. vinifera x V. berlandieri)*, 161-49C (*V. berlandieri x V. riparia*) and 110R (*V. rupestris* x *V. berlandieri*). Each rootstock was drip irrigated for six consecutive years (2012-2017) using two different deficit irrigation techniques: Regulated Deficit Irrigation (RDI) and Partial Root zone drying Irrigation (PRI). All combinations were irrigated with similar annual water volumes and application of the same designed deficit irrigation strategy (Table 1). The final goal of this DI strategy, with low water application and moderate water stress, was to increase water use efficiency (WUE) and to obtain very high-quality Monastrell grapes for premium red wine production. Crop evapotranspiration (ETc = ETo x Kc) was estimated using varying crop coefficients (Kc)—based on those proposed by the FAO, adjusted for the Mediterranean area—and reference evapotranspiration (ETo) values [14]. The ETo was calculated weekly from the mean values of the preceding 12–15 years using the FAO Penman–Monteith method [29] and the daily climatic data collected in the meteorological station (Campbell mod. CR 10X), located at the experimental vineyard and belonging to the Servicio de Información Agraria de Murcia (SIAM, IMIDA). The experimental design consisted of four replicates per rootstock–irrigation combination in a completely randomized 4-block design. Each replicate contained five vines, with only the three central vines being assessed; the border vines in each row were excluded to eliminate potential edge effects. Soil, water and plant characteristics, climatic factors, experimental conditions, ETo and Kc applied, and fertilizers used were described previously in detail [14].

**Table 1.** Deficit irrigation techniques, strategy, and water volume applied for each irrigation method (regulated deficit irrigation (RDI) and partial root-zone irrigation (PRI)) in each phenological period during the experimental period (2012–2017).

| Year | Irrig. Method | Budburst-Fruitset (mm) | Fruit Set-Veraison (mm) | Veraison-Harvest (mm) | Postharvest (mm) | Total Annual Water Volume Applied (mm year$^{-1}$) |
|---|---|---|---|---|---|---|
| | | April-May | June-July | Beginning of August-mid September | mid-September-end October | |
| | | % ETc (10−20) | %ETc (10) | %ETc (25−30) | %ETc (20−30) | |
| Average | PRI | 20.3 | 25.4 | 36.6 | 10.0 | 92.3 |
| (2012-2017) | RDI | 19.4 | 25.4 | 35.6 | 10.0 | 90.4 |

Each year at harvest, the yield (kg·vine$^{-1}$) was measured in 24 vines per rootstock (12 vines per irrigation method), and productive WUE (WUE$_{yield}$, Kg m$^{-3}$ applied) was calculated. The total berry quality index (technological and phenolic quality) (QI$_{overall\ berry}$) was calculated in Monastrell grapevines, with some modifications [14,30]. The harvest date was in accordance with the grower´s practice in the area, when °Brix reached 23.5−24.0. Between 40 and 50 kg of healthy grapes were collected for each combination (R x IM) to perform the microvinifications (3 per combination of R x IM) in 2014, 2015, and 2016. The wine quality index (QI$_{wine}$) after alcoholic fermentation was calculated as previously described [30].

### 2.2. Cost/Benefit Analysis and Productive, Economic, and Social Efficiency of Irrigation Water

To study the economic feasibility and profitability of these long-term deficit irrigation strategies, we used a cost/benefit analysis to calculate certain economic indexes [18]. The parameters and indexes used were: Net Margin/operating cost (NM/c) (%), NM/investment (NM/K) (%), NM/total cost (MN/C) (%), the average cost of production (€ kg$^{-1}$), and break-even point or viability threshold (kg ha$^{-1}$). The break-even point indicates the minimum quantity needed (kg ha$^{-1}$) from which the operation begins to generate positive results (Net Margin = 0).

Other indexes that are devoted to the analysis of the socio-economic efficiency of irrigation water were also calculated, due to the importance of this resource in the southeast of Spain. These indices were: Water productivity (€ m$^{-3}$) or Income per m$^3$ as an indicator of the gross income generated by each m$^3$ applied; Economic efficiency (€·m$^{-3}$) as the Net Margin generated by a m$^3$ of water, equivalent to a profit per m$^3$ [23]; and the productive water use efficiency (WUE) (kg·m$^{-3}$) as an indicator of kg of grapes produced by each m$^3$ applied in the crop. We also analyzed the social importance of each treatment according to the level of employment per cultivated hectare (Agricultural Work Unit, AWU·ha$^{-1}$) and cubic hectometer (AWU·hm$^{-3}$), respectively. Finally, we calculated the maximum price of irrigation water compatible with the economic viability of the activity (Water Viability Threshold, WVT, € m$^{-3}$), i.e., the price at which income and costs are equal [23]. Costs and income were the average of the six years of the trial, so they are representative of one production year. All cultivation practices were the same in all the treatments, with the exception of the differentials, i.e., irrigation and its associated energy needs, and pruning and gathering during winter. Such winter pruning and gathering was taken into account to establish the cost involved in each treatment.

In relation to the fixed costs, we calculated the annual depreciation costs (Table 2). In the fixed asset costs, all depreciations are the same, except for the irrigation network in the case of PRI with its double row of drippers. The initial investment of a holding of 10 hectares, as well as the depreciation of each item, was calculated by the linear or constant quotas method. The useful life was estimated based on the experience of the last years of the agricultural companies in similar crops, such as a real mean life. Finally, we showed the cost impact per hectare (Table 2).

**Table 2.** Investment and annual depreciation in Monastrell vineyard trellis systems.

| | Initial Value (€) | Residual Value (€) | Useful life (years) | Depreciation** (€/year) | Depreciation (€/ha) |
|---|---|---|---|---|---|
| Shed for equipment and irrigation control | 7200 | 1800 | 30 | 183 | 18 |
| Irrigation equipment | 7000 | 0 | 15 | 474 | 47 |
| Irrigation network* | 23,910 | 0 | 10 | 2427 | 243 |
| Planting | 76,660 | 0 | 25 | 3112 | 311 |
| Various | 200 | 0 | 10 | 41 | 4 |
| Irrigation Reservoir | 7400 | 1850 | 30 | 188 | 19 |
| Investment (€ ha$^{-1}$) | | | 12,237 | | |

*Investment in PRI irrigation networks are 4161 € ha$^{-1}$ and total investment is 14,007 € ha$^{-1}$. ** Annual depreciation plus opportunity cost (interest rate 1.5%).

The labor employed in different tasks, including operating machinery, was calculated to determine the employment generated. In the Region of Murcia, one Agricultural Work Unit (240 work days) corresponds to a total of 1920 hours.

Water is a variable cost—a function of the quantity applied and the established price. The prices of the years 2012–2017 of this resource are shown in Table 3. Income was calculated from the annual average sale price of Monastrell grapes in the 2012–2017 period in the Region of Murcia, obtained from the Statistical Service of the Ministry of Water and Agriculture and the Rosario de Bullas cooperative (Murcia). The latter is the largest winery by volume of production of PDO Bullas, representing almost 50% of the total production of wine. Mean income was calculated from the production, the prices paid in euros (€) per kilogram, the °Baumé (Table 3), and the average data for °Baumé in each treatment and year [14].

Thus, the income from each rootstock–irrigation strategy was calculated for each year, and the mean for the period 2012–2017 was used to establish mean income. Data for the calculation of income and costs for the period, as already indicated, were taken from previous physiological and agronomic studies [14,15], and were intended to show the structure of income and costs of an average representative year.

**Table 3.** Annual prices and average price of water and grapes for the period 2012-2017.

| Prices | 2012 | 2013 | 2014 | 2015 | 2016 | 2017 |
|---|---|---|---|---|---|---|
| Irrigation water (€ $m^{-3}$) | 0.19 | 0.19 | 0.20 | 0.20 | 0.20 | 0.22 |
| Grapes (€/kg ºBe)* | 0.0260 | 0.0225 | 0.0255 | 0.0220 | 0.0265 | 0.0300 |

\* Average price of Monastrell grape paid to the vine grower.

## 3. Results

The average income for each rootstock–irrigation method combination showed that the more productive rootstock (140Ru) had the highest income, while the lowest productive rootstock (161-49C) had the lowest income (Table 4).

**Table 4.** Average income (2012–2017) for each rootstock (R)–irrigation method (IM) combination.

| | 140Ru | | 161-49C | | 110R | | 1103P | | 41B | |
|---|---|---|---|---|---|---|---|---|---|---|
| | PRI | RDI | PRI | RDI | PRI | RDI | PRI | RDI | PRI | RDI |
| Yield (kg ha$^{-1}$) | 16,198 | 16,354 | 7098 | 8606 | 9802 | 8060 | 9932 | 9828 | 9802 | 10,010 |
| ºBaumé | 13.17 | 13.25 | 13.39 | 13.37 | 13.24 | 13.31 | 12.83 | 13.18 | 12.90 | 13.10 |
| Average grape price (€ kg$^{-1}$) | 0.329 | 0.331 | 0.327 | 0.327 | 0.325 | 0.328 | 0.321 | 0.332 | 0.318 | 0.322 |
| Total income (€ ton.$^{-1}$) | 329 | 331 | 326 | 327 | 325 | 328 | 320 | 332 | 318 | 322 |
| Total income (€ ha$^{-1}$) | 5332 | 5416 | 2320 | 2816 | 3182 | 2647 | 3183 | 3263 | 3117 | 3225 |

Taking into account the cost accounting of each combination of rootstock–IM (irrigation method) (Table 5), in general, the behavior of the 140Ru rootstock differed from the others in terms of productivity and vigor, which influenced the income and costs. There were two clearly differentiated groups, namely, 140Ru and the rest, since the operating cost per hectare of 140Ru was higher than the rest, all of which had similar costs (Table 5).

**Table 5.** Cost accounting for all combinations (R x IM) during the experimental period 2012–2017.

| | 140Ru | | 161-49C | | 110R | | 1103P | | 41B | |
|---|---|---|---|---|---|---|---|---|---|---|
| | PRI (€) | RDI (€) | PRI (€) | RDI (€) | PRI (€) | RDI (€) | PRI (€) | RDI (€) | PRI (€) | RDI (€) |
| Shed | 18 | 18 | 18 | 18 | 18 | 18 | 18 | 18 | 18 | 18 |
| Irrigation equipment | 47 | 47 | 47 | 47 | 47 | 47 | 47 | 47 | 47 | 47 |
| Irrigation network | 422 | 243 | 422 | 243 | 422 | 243 | 422 | 243 | 422 | 243 |
| Planting | 311 | 311 | 311 | 311 | 311 | 311 | 311 | 311 | 311 | 311 |
| Various | 4 | 4 | 4 | 4 | 4 | 4 | 4 | 4 | 4 | 4 |
| Regulator reservoir | 19 | 19 | 19 | 19 | 19 | 19 | 19 | 19 | 19 | 19 |
| Fixed assets | 822 | 642 | 822 | 642 | 822 | 642 | 822 | 642 | 822 | 642 |
| Annual pruning | 437 | 500 | 251 | 255 | 241 | 191 | 344 | 322 | 258 | 255 |
| Summer pruning | 206 | 206 | 206 | 206 | 206 | 206 | 206 | 206 | 206 | 206 |
| Machinery | 580 | 582 | 469 | 487 | 502 | 481 | 503 | 502 | 502 | 504 |
| Phytosanitary products | 106 | 106 | 106 | 106 | 106 | 106 | 106 | 106 | 106 | 106 |
| Fertilizers | 156 | 156 | 156 | 156 | 156 | 156 | 156 | 156 | 156 | 156 |
| Herbicides | 30 | 30 | 30 | 30 | 30 | 30 | 30 | 30 | 30 | 30 |
| Electricity | 16 | 15 | 16 | 15 | 16 | 15 | 16 | 15 | 16 | 15 |
| Harvesting | 1057 | 1068 | 462 | 562 | 639 | 525 | 648 | 641 | 639 | 653 |
| Irrigation | 187 | 183 | 187 | 183 | 187 | 183 | 187 | 183 | 187 | 183 |
| Operating costs | 2775 | 2847 | 1884 | 2002 | 2084 | 1894 | 2197 | 2163 | 2100 | 2111 |
| Total costs* | 3597 | 3489 | 2706 | 2644 | 2906 | 2537 | 3019 | 2805 | 2922 | 2753 |

\* Production cost per hectare.

The difference in vigor was reflected in the difference in the average cost of pruning in the analyzed period (2012–2017) (sum of annual and summer pruning in Table 5). While for the 140Ru, this amounted to 674 €·ha$^{-1}$·year$^{-1}$, in others, it was around 450–500 €·ha$^{-1}$·year$^{-1}$, a figure only surpassed slightly by 1103P (539 €·ha$^{-1}$·year$^{-1}$). In addition, 140Ru differed with regard to its productivity, achieving an average cost of production (Table 6) of 0.23 €·kg$^{-1}$, which was significantly below that of the rest of the rootstocks (it was followed by 1103P and 41B, 0.31 €·kg$^{-1}$ in both cases, and 110R with 0.32 €·kg$^{-1}$). In contrast, 161-49C rootstock showed a higher production cost (0.39 €·kg$^{-1}$, Table 6), mainly due to its lower productivity (Table 4).

Fixed assets were more linked to the installation of irrigation (60% of the fixed assets costs, Table 5). In the case of 140Ru, due to its high productivity, the cost of fixed assets was lower in relative terms (23%) compared to the other rootstocks. In contrast, in 161-49C, the lower productivity was also penalized with 30% of the cost of fixed assets (Table 5).

Among the operating costs, those associated with pruning and harvesting represented between 35%–40% of the total cost and more than 50% of the total operating cost. The operating cost related to harvesting (the most important cost) ranged from 38% of the total of operating costs for 140Ru to 26% for 161-49C (Table 5). The economic and efficiency indices such as Net margin/Total cost (%), Net margin/operating cost (%), NM/investment (%), break-even point (kg ha$^{-1}$), WUE (kg m$^{-3}$), water productivity (€ m$^{-3}$), and economic efficiency (€ m$^{-3}$) were significantly higher in 140Ru compared to the other rootstocks, and the lowest (negative values for NM/C, NM/c, NM/K, and economic efficiency; not viable economically) for 161-49C, while other rootstocks showed intermediate positive values of these economic/efficiency indexes (Table 6). In relation to the social importance of the crop (Table 6), the results indicate that the most vigorous and productive rootstock (140Ru) generated more employment (0.16 UTA/ha) and social efficiency (AWU hm$^{-3}$) and had a significantly higher WVT (2.16 € m$^{-3}$) (water price in which income and costs are equal) compared to the other rootstocks, due to an increased labor requirement and cost of pruning and harvesting. In contrast, less productive rootstocks (161-49C and 110R) generated less employment and significantly lower social efficiency and WVT (Table 6).

**Table 6.** Productive, economic and social parameters calculated in the cost/benefit analysis for different rootstocks (R), irrigation methods (IM) and the interaction (R x IM) for the period 2012–2017.

| | NM/Cost (%) | NM/Operating Cost (%) | NM/ Investment (%) | Average Cost (€ kg⁻¹) | Break-Even Point (kg ha⁻¹) | WUE (kg m⁻³) | Water Productivity (€ m⁻³) | Economic Efficiency (€ m⁻³) | Social Efficiency (AWU hm⁻³) | Employment (AWU ha⁻¹) | WVT (€ m⁻³) |
|---|---|---|---|---|---|---|---|---|---|---|---|
| **Rootstock (R)** | | | | | | | | | | | |
| 140Ru | 50.75c | 63.65c | 14.07c | 0.23a | 10,846c | 17.81d | 5.85d | 1.96c | 180d | 0.16c | 2.16c |
| 1103P | 9.73b | 12.26b | 2.46b | 0.31b | 9032b | 10.80c | 3.51c | 0.31b | 132c | 0.12b | 0.51b |
| 41B | 10.95b | 14.09b | 2.62b | 0.31b | 8846b | 10.90c | 3.48bc | 0.36b | 126b | 0.11a | 0.56b |
| 110R | 6.52b | 8.61b | 1.54b | 0.32b | 8265a | 9.82b | 3.21b | 0.22b | 117a | 0.11a | 0.42b |
| 161-49C | −5.54a | −9.20a | −0.67a | 0.39c | 8101a | 8.64a | 2.82a | −0.12a | 114a | 0.10a | 0.08a |
| **Irrigation method (IM)** | | | | | | | | | | | |
| PRI | 10.33 | 12.77 | 2.88 | 0.33 | 9340 | 11.50 | 3.72 | 0.42 | 133 | 0.12 | 0.62 |
| RDI | 18.63 | 22.99 | 5.13 | 0.30 | 8696 | 11.69 | 3.83 | 0.67 | 135 | 0.12 | 0.87 |
| **Year** | | | | | | | | | | | |
| 2012 | 28.19d | 37.19d | 6.49d | 0.29b | 7852b | 11.58c | 4.26d | 0.96d | 137c | 0.120bc | 1.16d |
| 2013 | 26.61d | 33.67d | 7.14de | 0.21a | 12,474d | 18.11d | 4.84e | 1.05d | 170d | 0.150e | 1.25d |
| 2014 | −17.57a | −26.01a | −2.99a | 0.47d | 7178a | 7.26a | 2.60a | −0.48a | 118a | 0.098a | −0.28a |
| 2015 | 0.19b | −0.52b | 0.50b | 0.28b | 10,916c | 12.25c | 3.35b | 0.07b | 137c | 0.125d | 0.27b |
| 2016 | 18.37c | 23.37c | 4.73c | 0.33c | 7734b | 10.06b | 3.71c | 0.66c | 124b | 0.116b | 0.86c |
| 2017 | 31.11d | 39.59d | 8.16e | 0.30bc | 7954b | 10.31b | 3.89c | 1.02d | 118a | 0.123cd | 1.22d |
| **Interaction (R x IM)** | | | | | | | | | | | |
| 140Ru PRI | 47.13d | 60.77e | 12.39i | 0.23a | 11,028j | 17.57i | 5.75e | 1.84e | 175e | 0.161f | 2.04e |
| RDI | 54.36d | 66.53e | 15.75j | 0.22a | 10,663i | 18.06j | 5.96e | 2.09e | 185f | 0.167f | 2.29e |
| 1103P PRI | 4.62b | 5.89b | 1.17c | 0.32bc | 9499h | 10.78f | 3.44cd | 0.15b | 132d | 0.121e | 0.35b |
| RDI | 14.83c | 18.62cd | 3.74g | 0.31bc | 8565e | 10.82g | 3.58d | 0.47cd | 132d | 0.119de | 0.67cd |
| 41B PRI | 5.77b | 7.52b | 1.39d | 0.32bc | 9152g | 10.71e | 3.39cd | 0.21cd | 125bc | 0.114cd | 0.41bcd |
| RDI | 16.14c | 20.67d | 3.86h | 0.30b | 8539d | 11.09h | 3.56d | 0.51d | 128cd | 0.115cd | 0.71d |
| 110R PRI | 10.05bc | 13.77bcd | 2.19f | 0.31bc | 8850f | 10.69d | 3.49cd | 0.32bcd | 124bc | 0.113bc | 0.52bcd |
| RDI | 2.99b | 3.44b | 0.90b | 0.34c | 7680a | 8.95b | 2.93b | 0.12b | 111a | 0.100a | 0.32b |
| 161-49C PRI | −15.92a | −24.08a | −2.75a | 0.44d | 8170c | 7.74a | 2.53a | −0.43a | 109a | 0.099a | −0.23a |
| RDI | 4.85b | 5.69b | 1.41e | 0.34c | 8032b | 9.54c | 3.11bc | 0.18bc | 120b | 0.108b | 0.38bc |
| **ANOVA** | | | | | | | | | | | |
| R | *** | *** | *** | *** | *** | *** | *** | *** | *** | *** | *** |
| IM | *** | *** | *** | ** | *** | ns | ns | *** | ns | ns | *** |
| Year | *** | *** | *** | *** | *** | *** | *** | *** | *** | *** | *** |
| R x IM | ** | *** | ** | *** | ** | ** | ** | ** | *** | *** | ** |

'ns' not significant; *, **, and *** indicate significant differences at the 0.05, 0.01, and 0.001 levels of probability, respectively. In each column and for each factor, different letters indicate significant differences according to Duncan's multiple range test at the 95% confidence level..

The analysis of profitability based on grape price variability revealed that high berry quality rootstocks (high $QI_{overal\ berry}$ and $QI_{wine}$, Table 7) were not viable economically until the grape price rose up to 0.024 € kg° for 110R and 0.030 € kg$^{-1}$ °Be$^{-1}$ for 161-49C (Figure 1). In contrast, in low berry quality rootstocks such as 140Ru and 1103P (lower $QI_s$, Table 7), viability and economic profitability were obtained with lower grape prices (0.020 € kg$^{-1}$ °Be$^{-1}$ for 140Ru, and 0.024 € kg$^{-1}$ °Be$^{-1}$ for 1103P). In addition, very productive combinations (rootstocks–IM) such as 140Ru PRI and RDI allowed lower grape prices (from 0.016 € kg$^{-1}$ °Be$^{-1}$) to be viable economically and to obtain high profitability (Figure 2). In contrast, for low productive combinations of rootstocks–IM (especially 161-49C PRI, with the highest berry/wine quality, Table 7), we needed to increase grape price to almost double (above 0.030 € kg$^{-1}$ °Be$^{-1}$) to start getting an economic return for the grower (Figure 2).

**Table 7.** Overall berry quality index (QI $_{overall\ berry}$) calculated for Monastrell grapes at harvest for five different rootstocks (140Ru, 1103P, 41B, 110R, and 161-49C) and two different irrigation methods (PRI and RDI) from 2012 to 2016. Wine quality index $QI_{wine}$ after alcoholic fermentation calculated for Monastrell for five different rootstocks (140Ru, 1103P, 41B, 110R, and 161-49C) and two different irrigation methods (PRI and RDI) from 2014 to 2016.

| Rootstock (R) | | QI $_{overall\ berry}$ | $QI_{wine}$ |
|---|---|---|---|
| 140Ru | | 9.8a | 1.56b |
| 1103P | | 10.0a | 1.62b |
| 41B | | 10.8b | 1.38a |
| 110R | | 11.2b | 1.80c |
| 161-49C | | 12.3c | 1.83c |
| **Irrigation method (IM)** | | | |
| PRI | | 11.2 | 1.68 |
| RDI | | 10.5 | 1.60 |
| Year | | | |
| 2012 | | 12.6d | - |
| 2013 | | 7.7a | - |
| 2014 | | 10.6b | 2.33c |
| 2015 | | 11.7c | 1.77b |
| 2016 | | 11.5c | 0.83a |
| **Interaction (R x IM)** | | | |
| 140Ru | PRI | 10.2bc | 1.45abc |
| | RDI | 9.4a | 1.67bc |
| 1103P | PRI | 10.2bc | 1.49abc |
| | RDI | 9.8ab | 1.75c |
| 41B | PRI | 10.7cd | 1.24a |
| | RDI | 10.8cd | 1.51abc |
| 110R | PRI | 11.2d | 1.83c |
| | RDI | 11.3d | 1.77c |
| 161-49C | PRI | 13.5e | 2.39d |
| | RDI | 11.1d | 1.28ab |
| **ANOVA** | | | |
| R | | *** | * |
| IM | | *** | ns |
| Year | | *** | *** |
| R x IM | | *** | *** |

ns, not significant; *, **, and *** indicate significant differences at the 0.05, 0.01, and 0.001 levels of probability, respectively. In each column and for each factor or interaction, different letters indicate significant differences according to Duncan's multiple range test at the 95% confidence level.

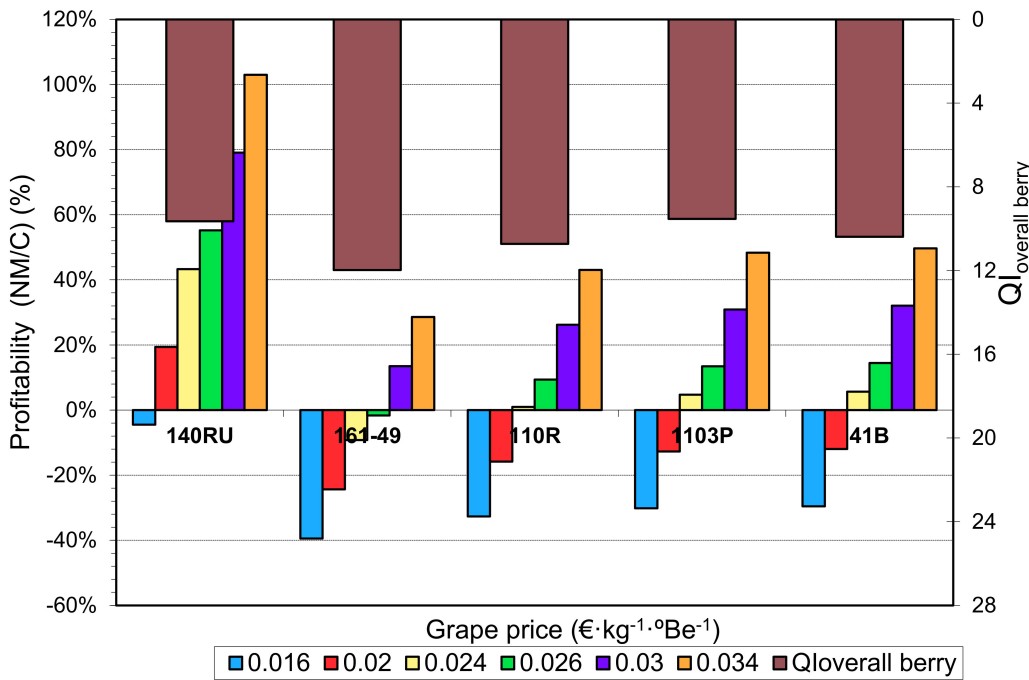

**Figure 1.** Profitability (NM/C, %) for each rootstock based on grape price variability (€ kg$^{-1}$ °Be$^{-1}$) for the period 2012–2017 in a Monastrell vineyard in southeastern Spain. Average of the values of QI$_{overall\ berry}$ for each rootstock for the period (2012–2017).

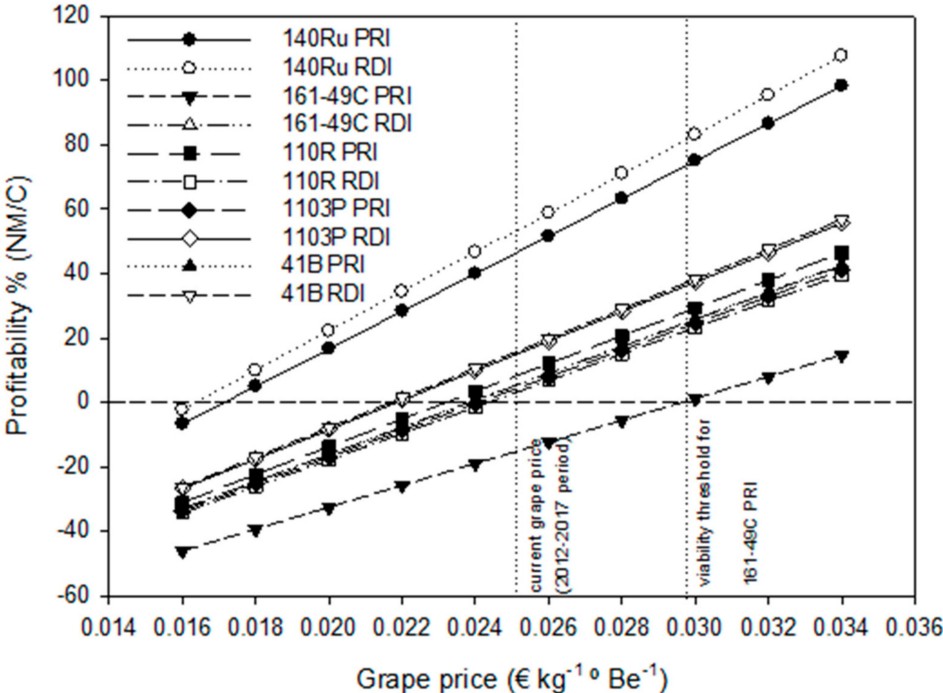

**Figure 2.** Profitability (NM/C, %) based on grape price variability (€ kg$^{-1}$ °Be$^{-1}$) for the period 2012–2017 for each combination of rootstock–IM in a Monastrell vineyard in southeastern Spain. The vertical dotted lines represent the weighted average market grape price (0.0254 € kg$^{-1}$ °Be$^{-1}$) for the period 2012–2017 and the price of grapes necessary to reach the viability threshold (B/C = 0) of 161-49C PRI (0.0296 € kg$^{-1}$ °Be$^{-1}$), the most unfavorable combination. The horizontal short dashed line represents the viability threshold (B/C = 0).

The analysis of profitability based on the variability of prices of irrigation water also revealed that the most productive combinations (140Ru PRI and RDI) remained very profitable economically (above 40%), even with very high water prices (up to 0.40 € m$^{-3}$), compared to the other combinations (Figure 3). In contrast, the 161-49C PRI combination was not viable, neither with the current price of irrigation water nor with the increase in the price of water.

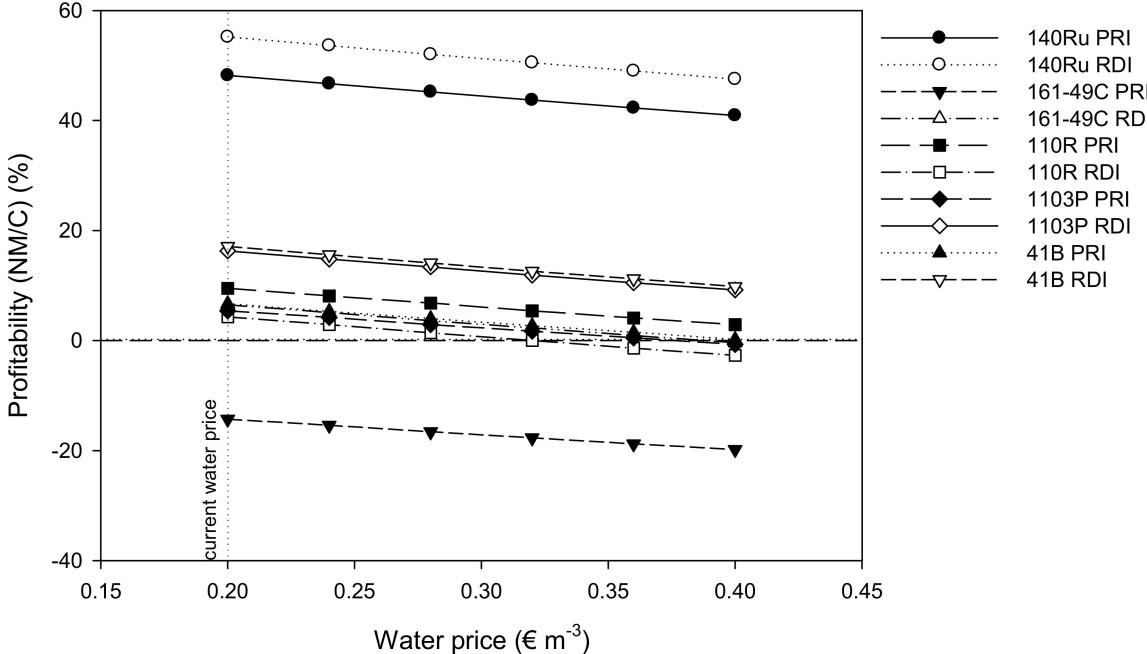

**Figure 3.** Profitability (NM/C, %) based on water price (€ m$^{-3}$) variability for each combination of rootstock–IM during the period 2012–2017 in a Monastrell vineyard in southeastern Spain. Vertical dotted line represents the current averaged price of irrigation water for the period (2012–2017) in the Murcia Region, southeastern Spain.

The analysis of the relationships between the efficiency ratios, economic indices, and berry quality index showed that high WUE was closely related with high economic efficiency and break-even point, according to a significant positive relationship (Figure 4A,B). In contrast, QI$_{overall\ berry}$ was inversely related with break-even point, yield, water productivity and economic efficiency (Figure 4C–F). In addition, QI$_{overall\ berry}$ was positively associated with production costs (Figure 4G). Production costs were also related in an exponentially decayed way with WUE$_{yield}$, while QI$_{overall\ berry}$ was inversely and linearly related with WUE$_{yield}$ (Figure 5).

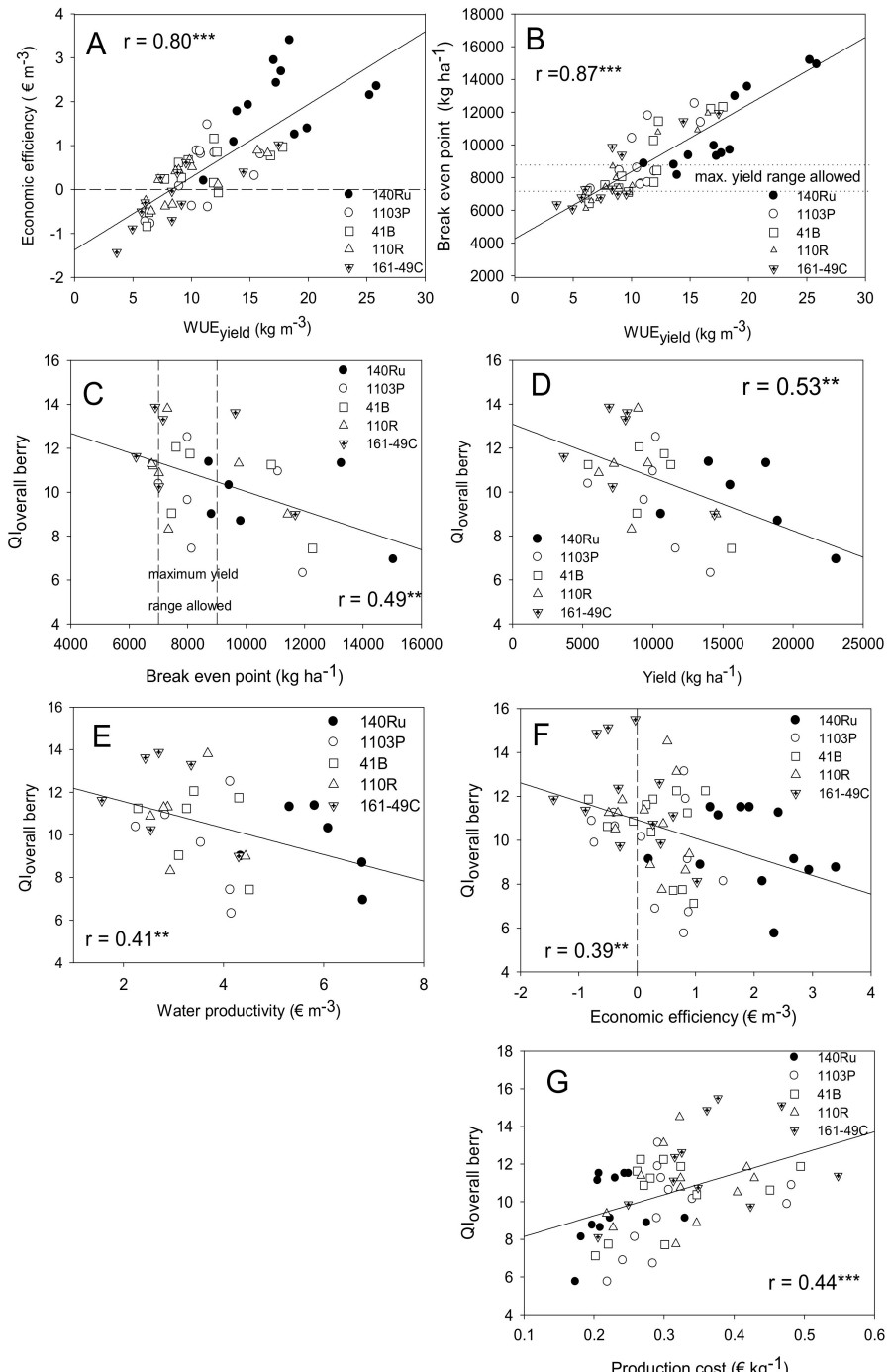

**Figure 4.** (**A**) Significant relationship between economic efficiency and $WUE_{yield}$, (economic efficiency = -1.3739 + 0.1656* $WUE_{yield}$). (**B**) Significant relationship between break-even point and $WUE_{yield}$ (Break-even point = 4261.62 + 410.1788 $WUE_{yield}$) and (**C**) between $QI_{overall\ berry}$ and break-even point ($QI_{overall\ berry}$ = 14.4451 – 0.0004* break-even point). (**D**) Significant relationships between $QI_{overall\ berry}$ and yield ($QI_{overall\ berry}$ = 13.0909 – 0.0002* yield), (**E**) between $QI_{overall\ berry}$ and water productivity ($QI_{overall\ berry}$ = 12.8174 – 0.6236* water productivity) and (**F**) between $QI_{overall\ berry}$ and economic efficiency ($QI_{overall\ berry}$ = 10.9249 – 0.8447* economic efficiency). (**G**) Significant relationships between $QI_{overallberry}$ and production costs ($QI_{overallberry}$ = 7.0317 + 11.1401* productions costs). For each rootstock, each single point represents the average per year and irrigation method (period 2012–2017). Dashed lines in A and F indicate when economic efficiency is = 0 (not viable economically). Dashed lines in B and C indicate maximum yield range allowed for Monastrell red berries in O.D. Bullas, SE Spain.

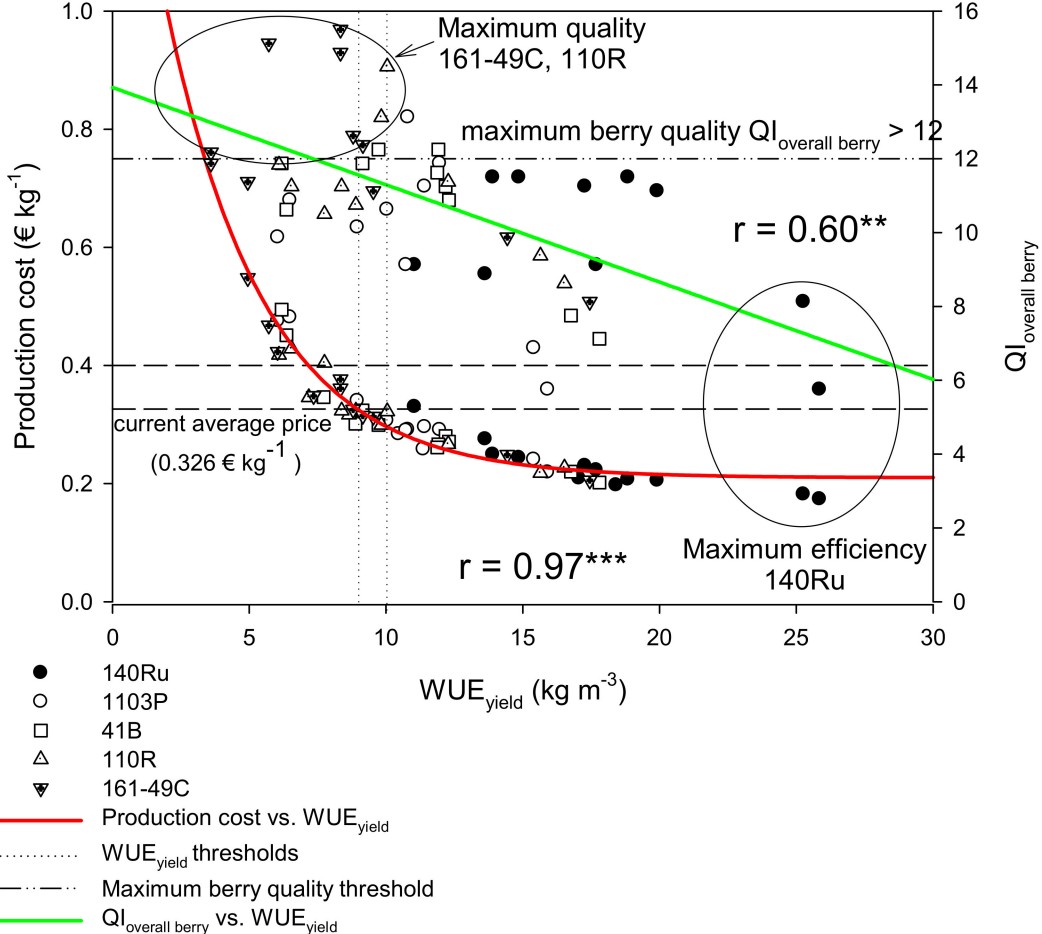

**Figure 5.** Significant relationship between production cost and $WUE_{yield}$ (Production costs = 0.21 + 1.3714* $e^{(-0.2757*WUEyield)}$ and between $QI_{overall\ berry}$ and $WUE_{yield}$ ($QI_{overall\ berry}$ = 13.9310 − 0.2636* $WUE_{yield}$). For each rootstock, each single point is the average per year and the irrigation method (period 2012–2016).

## 4. Discussion

Monastrell grafted on all rootstocks were economically viable crops, with the exception of 161-49C, in the current grape market conditions. Vines grafted on 140Ru and 1103P were the most productive, providing the best economic results and the highest WUE (kg m$^{-3}$), but they showed low grape and wine quality indexes (Table 7). The greatest profitability was reached with 140Ru (NM/C = 50.75%), mainly due to increased vigor and productivity (kg·ha$^{-1}$), because there was practically no difference in °Baumé, (around 13 °Baumé in all rootstocks). On the contrary, vines grafted on rootstocks 161-49C and 110R were the least productive and vigorous [14], but had significantly increased grape and wine quality (Table 7). 110R rootstock was economically viable but showed low profitability (6.52%), while the 161-49C was not viable with a negative NM/C ratio (%) and the lowest WUE (kg m$^{-3}$) and social efficiency (AWU hm$^{-3}$), indicating that the cost of producing grapes with this low vigor/productive rootstock in these irrigation conditions and with the current low prices of the grapes surpassed the income obtained.

All rootstocks, except 140Ru, had a break-even point (kg ha$^{-1}$) (Table 6) close to the maximum permitted by the regional PDO for grapes used for QWpsr wines. The rootstocks that obtained the highest quality grapes (161-49C and 110R) had a break-even point of around 8000 kg·ha$^{-1}$, which is within the limit set by the PDOs of southeast Spain. Therefore, in terms of yield/quality, they reached optimal values if destined for QWpsr wines. However, vines grafted on 140Ru had a higher break-even

point (almost 11,000 kg ha$^{-1}$ year$^{-1}$) and annual yields of 16,000 kg ha$^{-1}$, exceeding the limits established by PDO. All efficiency indicators (yield, WUE$_{yield}$, economic and social efficiencies, AWU values per hm$^{-3}$ and ha$^{-1}$) showed that Monastrell vineyards grafted on 140Ru had significantly higher efficiency and also generated significantly more profitability and employment compared to other rootstocks (Table 6). It is particularly noteworthy that the productive WUE efficiency reported in 140Ru vines (17.81 kg·m$^{-3}$) was very high for DI wine grapes in semiarid areas [12]. Besides, the high gross water productivity for this crop (around 3–6 €·m$^{-3}$ in all cases, which rose to 5.85 €·m$^{-3}$ in 140Ru) was very high in comparison with other wine-growing regions—such as Brasil (1.17 € m$^{-3}$) [31] and the Guadiana river basin (Spain) (1– 3 € m$^{-3}$) [32]—and with other horticultural crops (onions 2.96 € m$^{-3}$, potatoes 2.03 € m$^{-3}$, carrots 1.62 € m$^{-3}$; [33] and cereals (0.77-1.01 € m$^{-3}$) [34].

Thus, this vigorous, productive, efficient, and drought-tolerant rootstock (140Ru) could be adapted to more restrictive deficit irrigation strategies in semiarid areas, employing a lower volume of water, or even in rainfed conditions, in order to control the excess vigor and yield and to further enhance WUE and berry/wine quality. Alternatively, the use of 140Ru could also be a good alternative, especially for the preparation of other types of wine (table wines), not limited in production by quality standards.

Among the operating costs, those associated with pruning and harvesting represented between 35%–40% of the total cost and more than 50% of the total operating cost. This crop had a major social impact as a generator of rural employment, since the operating cost is more associated with manual tasks than consumable factors of production (fertilizers, pesticides, etc.). In general, social efficiency values in vineyards of southeastern Spain were better than other crops (stone fruits, pome fruits, citrus, etc.) [21,23,35]. In relation to the social importance of the crop (Table 6), we reported similar higher values for overall employment (between 0.10 and 0.16 AWU·ha$^{-1}$) than those obtained for vineyards growing on trellises in different locations (0.13 AWU ha$^{-1}$ and 0.10 AWU·ha$^{-1}$) [17,36]. 140Ru stood apart at 0.16 AWU·ha$^{-1}$, due primarily to the increased productivity and vigor and the higher operating (labor) costs (increased cost of pruning and harvesting). The values obtained are consistent with the average for the European Union (0.12 AWU·ha$^{-1}$) and are more than double those recorded for agricultural holdings as a whole (0.05 AWU·ha$^{-1}$) [37]. These indicators confirm the value of DI vineyards as a very important crop, being socio-economic motors for territories, closely linked to the environment and rural development in arid and semiarid areas, in which, in many cases, there are not many productive possibilities (because of very limited water resources or climatic and soil limitations). We have only referred to the phase of cultivation, but the subsequent phases of processing and marketing of QWpsr wines increase the socioeconomic importance of this crop.

The water viability threshold indicated that four rootstocks (140Ru, 1103P, 41B, and 110R) were adapted to the existing prices of water and, even, in the case of 140Ru, very high prices of water of up to 2 € m$^{-3}$ could be supported. Only 161-49C, due to its lower productivity, showed a lower threshold than the existing price of water and is, therefore, not viable under current conditions.

In general, RDI strategies were better than PRI strategies in economic terms (economic efficiency and WVT) in practically all rootstocks, except in 110R (where PRI was more beneficial than RDI) (Table 6). This advantage of RDI may be due to two reasons: the first, that the fixed cost of PRI strategies represented a cost greater than RDI due to the dual network of irrigation—in particular, the PRI treatments cost 180 € ha$^{-1}$ per year more than RDI; and a second cause, the gross income (€ ha$^{-1}$) of RDI was higher in all rootstocks, except in 110R (Table 4). It is interesting to note that the 161-49C rootstock, which was not viable in global terms (taking into account the average of the rootstock, including both PRI and RDI), was viable in RDI conditions (Table 6). It is likely that excessive water stress caused by PRI in this rootstock, because of the low volumes of irrigation applied in the wet root zone, strongly affected its productivity and, therefore, its profitability, although the technological quality and grape polyphenols were clearly improved [14]. These results suggest that the implementation of PRI could be improved in this unproductive rootstock by increasing annual irrigation volumes, an aspect that needs to be investigated. Other combinations like 161-49C RDI or 110R PRI may be good strategies for

use in arid conditions since they are profitable and more productive, while maintaining a good quality of grape and wine (Table 7).

We would expect that a price premium for a certain wine variety or appellation would translate into a price premium for the corresponding wine grape variety and grape location, but this is not always the case. In recent years, there has been increasing concern in the wine grape market for the need to establish protocols and methods to classify different qualities of grapes intended to make quality wines [18,28]. Our group has developed some berry and wine quality indices (based on technological and phenolic quality) that help in the differentiation of Monastrell grapes and wines in relation to each tested strategy or combination [30]. The possibility of increasing the price of grapes through a premium on quality would change all of the profitability scenarios shown in this work, especially considering that most Spanish wine consumers are willing to pay a price premium for a greater quality and more sustainable wine, and that there are differences among the main market segments [38]. Thus, our analysis showed that 161-49C was the only rootstock that was unviable at the average current price of the period (Figure 1). However, with a premium price of 0.030 €·kg$^{-1}$ °Be$^{-1}$, equivalent to 0.40 €·kg$^{-1}$, this option was cost effective (NM/Cost = 13.5%) (Figure 1). That is, only with a premium of 6 cents, the use of rootstock 161-49C would be profitable for Monastrell production. In addition, this rootstock produced higher quality grapes and wines (Table 7) and was at the limit marked by the majority of PDO (Origin Denominations) in the Spanish Mediterranean area (7000–9000 kg·ha$^{-1}$). Besides, the evolution of the NM/C indicator with the price of the grape indicated that all combinations of rootstock–IM were viable with the weighted average market price of the period 2012–2017 (0.0254 €·kg$^{-1}$ °Be$^{-1}$), except the one with the highest quality, i.e., 161-49C PRI (Table 7, Figure 2). These grapes, which potentially give a better wine (Table 7), should have a premium set for their high quality to make it economically viable (NM/C = 0). In particular, if the average price paid had been just 0.0296 €·kg$^{-1}$ °Be$^{-1}$ (about 5 cents per kg more than the current price), which is equivalent to an increase in the price of 16.5%, 161-49C PRI would have been viable, while the rest of the combinations that also gave higher berry and wine quality, such as 161-49C RDI, 110R PRI, and 110R RDI would have provided a more than 20% increase, which is a good economic choice for the Monastrell vines in these semiarid conditions (Figure 2). In Spain, the lowest grape prices correspond to traditional wine-growing areas of Southeastern Spain (Valencia, Alicante, Murcia) and Castilla la Mancha, and to varieties such as Monastrell, Bobal, and Cencibel (0.24 and 0.33 € kg$^{-1}$), while the highest grape prices (0.85–1.30 € kg$^{-1}$) were paid to varieties such as Tempranillo in the areas of North of Spain, Ribera del Duero, and Rioja [39]. It is noteworthy that these semiarid areas of Southeastern Spain (Valencia, Murcia) and Castilla la Mancha (with the lowest grape prices) will be more vulnerable to the effects of climate change than other wine-growing regions and will need the most effort to adapt, with increased costs to maintain the quality and productivity of vineyards, since these regions will face changes of greater magnitude than other wine-producing areas [6]. Thus, if the grape prices do not rise substantially and if sustainable adaptation measures are not taken quickly, wine grape production will likely disappear in these traditional wine-growing areas. For instance, in the region of Murcia, a reduction of 12,224 ha was already observed in the vineyard surface in the period 2009–2017 [40].

In contrast, the current averaged prices paid for red grapes in other wine-growing regions worldwide, in general, are also higher than those paid in southeastern Spain, depending on variety and wine-growing region, oscillating between 0.32 € kg$^{-1}$ in South African wine-growing regions [41] (with production costs and grape prices similar to this study), 0.29–1.43 € kg$^{-1}$ in different wine-growing regions in Australia [42], 1.21–1.57 € kg$^{-1}$ in Ontario (Canada) [43], 0.93–1.80 € kg$^{-1}$ in New Zealand [44], and up to 2.51–4.95 € kg$^{-1}$ in premium wine-growing regions such as Sonoma and Napa County (California, USA) [45].

On the other hand, in Spanish southeastern Mediterranean vineyards, production costs are also lower (around 3000 € ha$^{-1}$, Table 5) than in other wine-growing areas [46–48], due in part to the lower consumption of water, fertilizers, and phytosanitary products [13,18]. These vineyards, irrigated with highly efficient water-use DI strategies and low water volumes, are also more efficient

in the use of agrochemicals, with the relative cost of agrochemicals (fertilizers, phytosanitary and herbicides) between 8% and 10% of the total cost, compared to 20.4% in Ontario (Canada) [47], 13.2% in Russian River valley (California) [48], 13.6% in La Rioja (Spain) [49], and 11.6% in Murray Valley (Australia) [46]. This lower use of agrochemicals produces a lower environmental impact, since fertilizers and pesticides generate high pollutant emissions, both during their production and during their subsequent application in the field [50–53]. These factors contribute between 30% and 80% to the carbon footprint in viticulture [54]. In our area, in particular, they have calculated a global fertilizer and phytosanitary contribution to greenhouse gas emission of 78% [55].

In the semiarid wine-growing regions of southern Europe (such as southeastern Spain), climate models predict more frequent and longer periods of drought, an increase in temperatures, evaporative demand, and the water needs of vines [3,6]. For this, we considered it interesting to make a simulation for the foreseeable increases in the price of water, since the water resource for agricultural use will become increasingly more limited in semiarid areas. Therefore, the economic data indicate that in the not too distant future, the sustainability of vineyards will be seriously threatened in these areas due to higher water prices. For example, with a moderate rise in the price of water of between 12–18 cents, compared to the current price ($0.20\,\text{€·m}^{-3}$), i.e., reaching 0.32 or $0.38\,\text{€·m}^{-3}$, options such as RDI 110R or RDI 161-49C that until now have been profitable for PDO-permitted production because of their good quality grapes and wines, will be unviable economically, and the rest of the options will significantly diminish their profitability, with the most profitable ones becoming the more vigorous rootstocks 140Ru and 1103P (Figure 3).

The model shows that in a situation with high water prices, the best option to find a compromise between quality, production, and profitability for the grower would be a rootstock such as R110 using PRI, since it tolerates water prices of up to $0.52\,\text{€·m}^{-3}$ with high yields and good quality. The use of the PRI technique in 110R allows vineyards to increase or maintain berry and wine quality with an increase in yield and wine volume, compared to RDI [14,15], which can be a more profitable option.

If grape prices continue to be as low as the current ones, and if the grower is not rewarded for the quality of the grapes, only the productivity vision will continue and the cost-effective option will be to produce a lot of grapes, even if at the expense of berry and wine quality. There is therefore a clear conflict between productivity and quality in wine grape production. Our analysis shows that most of the productive and economic indices, such as yield, economic efficiency, break-even point, and water productivity, are inversely related to berry quality (Figure 4C–F). In addition, the relationships between $WUE_{yield}$, production costs, and $QI_{overall\ berry}$ indicate that in the current wine market conditions, maximum productive efficiency is closely related to low productive costs and $QI_{overall\ berry}$ (Figures 4 and 5). In contrast, maximum berry quality is closely related to lower $WUE_{yield}$ and higher production costs (Figure 5). Although the most vigorous and productive rootstocks (especially 140Ru, followed by 1103P) have higher absolute costs per ha (mainly due to the more intense manual labor in pruning and harvesting, Table 5), in terms of unit production costs (what it costs to produce a kilo of grapes, $\text{€ kg}^{-1}$), these rootstocks are more efficient (show lower unit production costs, meaning it costs less to produce a kg of grapes, Table 6) compared to low productive rootstocks. The analysis shows that in the very high $WUE_{yield}$ range, between 10 and $25\,\text{kg m}^{-3}$, production costs are quite low, even below the current average price, due to very high productivity, making this option very profitable economically overall in high vigor rootstocks such as 140Ru. In contrast, with lower $WUE_{yield}$ (between $5-10\,\text{kg m}^{-3}$), production costs start to increase sharply in an exponential way, whilst also increasing progressively the berry quality in a linear way (Figure 5). Below $5-6\,\text{kg m}^{-3}$, production costs increase a lot, with little effect on berry quality. In these conditions, the rapid increase in production costs due to very low productivity makes this option economically unfeasible. In this situation, it will be difficult to implement optimized deficit irrigation strategies and a sustainable irrigation water use, and the pressure on water resources will increase in semiarid areas.

According to the $WUE_{yield}$–production costs–$QI_{overall\ berry}$ relationship analysis, maximum $QI_{overall\ berry}$ ($\geq 12$) was reached at a $WUE_{yield}$ of around $7.3\,\text{Kg m}^{-3}$ (161-49C), which supposes

a low yield, around 7256 kg ha$^{-1}$ year$^{-1}$, similar to the yield range obtained with 161-49C PRI, and within the yield range allowed by O.D. Bullas. Unfortunately, with the current grape price (0.326 € kg$^{-1}$), this maximum quality option is unfeasible economically. In this situation, everything that supposes a production cost higher than the current grape price is not viable economically. According to our results, this corresponds with a WUE$_{yield}$ < 9 kg m$^{-3}$ and a yield < 8128 kg ha$^{-1}$. Thus, to maintain low production costs and high berry quality, the analysis aims for yield ranges between 8100 and 9000, thus not exceeding the range allowed for D.O. Bullas wines (yield range 7500–9000 kg ha$^{-1}$, and WUE$_{yield}$ between 9 and 10 kg m$^{-3}$), which, with the current price of the grapes, could be a good compromise between productivity, quality, and returns for the grower.

Taking into account the various problems that the wine sector faces, such as the decrease in the consumption of wine per capita, health and road safety problems associated with the consumption of alcohol, and strong competition with wines from other regions of the new world with less wine-making traditions, our results indicate that in Mediterranean semiarid areas, we must bet on quality as a differentiating character in wine production. That is, we should prize the typicality and the varietal character of wines from these Mediterranean regions; in the case of southeastern Spain, the Monastrell variety should be the focus of attention. This local ancient variety is well adapted to these harsh and dry climates of high temperatures and recurrent drought cycles, and possesses important wine-making potential, as evidenced by the wines of high quality and of great economic value currently produced with the Monastrell grape in different parts of the world [56]. Wines of this variety offer an alternative and very positive differentiating character compared with wines made from varieties more widely extended internationally. Therefore, we believe that public policies should encourage vine growers to invest in producing high-quality grapes (not only of technological quality, but also of polyphenolic and nutraceutical quality, and being chemical residue-free), as well as to develop agronomic practices in the vineyards that are environmentally and socially sustainable, by paying prices that are more adjusted to their current quality and real production costs. In this way, the production costs should take into account environmental and social costs, too. This fits with the most accepted concept of sustainability, which is defined through the three overlapping principles of environmentally sound, economically feasible, and socially equitable production [38]. Thus, the necessary changes that the industry will need to make over the coming years to remain competitive will be to introduce environmentally sustainable practices (e.g., to reduce inputs and implement organic and agroecological practices), to increase global grape and wine quality, and to maintain sustainable grape pricing, among other factors. With this low-input viticulture, we can implement agronomic measures such as optimized regulated deficit irrigation techniques with low water volumes, and we can use more efficient and drought-tolerant rootstocks that will improve efficiency in the use of water, fertilizers, and agrochemicals, which will improve the quality of the grapes and wine made in semiarid regions in a context of global warming and water-limiting conditions.

**Author Contributions:** Formal analysis, J.G.G.; supervision, J.G.G and P.R.A.; writing—original draft, P.R.A.; writing—review and editing, P.R.A. and J.G.G. All authors have read and agreed to the published version of the manuscript.

**Funding:** This work was financed by the Instituto Nacional de Investigación y Tecnología Agraria y Alimentaria (INIA), Subprograma Nacional de Recursos y Tecnologías Agrarias, en coordinación con las Comunidades Autónomas, through the Project RTA2012-00105-00-00, with the collaboration of the European Regional Development Fund and the project AGL2017-83738-C3-2-R (Ministerio de Ciencia, Innovación y Universidades).

**Acknowledgments:** The authors thank Francisco Javier Martínez López, for field assistance, and Eva María Arques Pardo for support in laboratory analyses. They also thank Philip Thomas for assistance with the manuscript preparation and correction of the written English.

**Conflicts of Interest:** The authors declare no conflict of interest.

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
