# Peer review of "The Productive, Economic, and Social Efficiency of Vineyards Using Combined Drought-Tolerant Rootstocks and Efficient Low Water Volume Deficit Irrigation Techniques under Mediterranean Semiarid Conditions"

_sustainability, doi:10.3390/su12051930_

Round 1

Reviewer 1 Report

Paper is highly original and shows WUE concerns under a very interesting point of view. Novel and useful information is provided concerning a common situation in many Mediterranean areas under grape cultivation.

In my opinion English style must be strongly improved. A number of paragraph are too large and dense, would be better more and shortened sentences. For instance, the last sentence of page 1 of introduction continuing with page 2 is very hard to follow. At page 3, the sentence starting from “It is necessary…...” and the following paragraph are not easy to understand. Many more parts along the text need to be reorganized.

Results are clear described by figures and appear consistent. However, in my opinion, the weakest point of the paper is in material and methods.

First, experimental design needs to be detailed in this paper, instead of advising the reading of a previous one (Romero et al. 2018).

Second, efforts should be spent in order to explain how measuring just 12 plants for each treatments is possible to extrapolate data referring to field conditions in terms of hectares. Maybe opportune statistical or modeling approaches need to be clarified. In a 0,4 Ha vineyard about 1100-1200 vines should be planted considering common density for SE vineyards. Therefore 110-120 vines for treatment are available, thus it is important to clarify the field design in this 0,4 Ha, the distribution of chosen plants belonging to each treatment, in order to infer the absence of possible interactions among rootstoks.

Third, the selection of plants and bunches for microvinifications needs more details.

Table 2 is neither mentioned in the text nor well explained.

I can’t understand what is 1,200 mm14 (page 4).

Discussion is very interesting and well set out. Maybe you could highlight that your results are apparently in contrast with previous studies (in particular I refer to the Medrano-Flexas group from Balearic Islands as weel as Chaves group from Portugal) in which WUE is generally positively related to fruit and must quality. This is just a suggestion, it could be interesting showing as a more biological-physiological approach led to different conclusions if compared to a more agronomic/market-based one.

The sentence at page 16 (Thus, this vigorous, efficient……………...berry/wine quality.) may appear something contradictory with your results

Reviewer 2 Report

General comments:

The manuscript presents results from a 6 year study that examined the effects of rootstock and deficit irrigation on productivity and fruit/wine quality and the associated economic implications. It follows an earlier publication from the same field trial that focused on the viticulture / physiology results, and it appears to be a well-designed and thoroughly conducted study. While possibly fortunate that the trial included 140Ru, the main points of the manuscript relating to production v higher quality with lower inputs is becoming increasingly important in many countries.

Where I do have comments on the manuscript a number of these are related to the calculated indices. Possibly some of these may be familiar to an economist, but coming from a more general viticulture background I found the relationships between some of these hard to follow in the figures. In particular, (and as per point raised in publication below), where you have a common parameter on both axis is the relationship meaningful? For example Figure 4A which has yield on both axis and very high correlations compared to 4H which has the sort of variability expected but highlights one of the key points of the study with higher quality associated with higher costs of production.

Kronmal, Richard A. "Spurious correlation and the fallacy of the ratio standard revisited." Journal of the Royal Statistical Society: Series A (Statistics in Society) 156.3 (1993): 379-392

The background and objectives raised in the introduction are clear and the discussion returns and summarises the outcomes and conclusion effectively.

Specific comments:

No line numbers in the version I received so I’ve just copied the relevant section of text across.

Page 2. Text ‘The increase in temperature will generate a water shortage at atmospheric level that will produce an increase in the rate of evapotranspiration, which some studies estimate will be 25% higher than the current rate by the end of the 21st century (Savé et al., 2017).

comment:

Savé et al. 2017 reference not very accessible for broader international audience. Maybe something like the following could be added?

Dezsi, Ştefan, et al. "High‐resolution projections of evapotranspiration and water availability for Europe under climate change." International Journal of Climatology 38.10 (2018): 3832-3841.

Page 3: PRI (Partial Root Drying Irrigation)

Comment: PRD would be a more commonly used / recognised acronym?

Page 3: Text ‘However, there is usually a …. Small improvements in the quality of rainfed grape or those grown under controlled deficit irrigation, together with the social and environmental importance of the wine industry in arid areas, make the activity viable and even profitable (García-García et al., 2008; García-García, 2016)’

Comment: The point being made in this paragraph and last sentence is not quite clear. By ‘the activity’, do you mean deficit irrigation or wine grape production? In the context of the study I would take it as deficit irrigation, but the sentence also includes improvements in the quality of rainfed grapes.

Page 4: Not a major point, but what was below 80 cm? In Romero et al 2018 the diviner tubes are described as being installed to 1 m and in the text (page 79) the deeper soil layers are given as 60-100 cm (although only 60-80 cm in the supplementary table). Was there a significant limitation to root growth at this depth?  

Page 4: 1,200 mm14. Annual ETo? Seems high for crop. What is the 14 superscript referring to?

Page 4: Except for 2017, table 1 provides the same information as Romero et al. 2018. Could just be reduced to the average? In the text it stated that all rootstocks received ‘similar’ annual water volumes. To clarify, each rootstock under PRI / PRD would then have received an average 92.3 mm and RDI 90.4 mm?

Page 5: Text ‘The Production efficiency of water (kg·m-3) or Water Use Efficiency (WUE) as an indicator of kg of grapes produced by each m3 consumed in the crop.’ Also related to same point WUE in Table 6 and text from page 17 ‘It is particularly noteworthy that the productive WUE efficiency reported in 140Ru vines (17.81 kg·m-3), was the highest found in the literature for DI wine grapes as far as we know.’

Comment: Some clarification is required regarding rainfall and irrigation. From the economic comparison just irrigation is the reasonable comparison, but when comparing to other studies (also needs reference) then the amount of rainfall during the season could influence the values as a higher rainfall season may have used less irrigation and therefore appear to have better WUE based on irrigation alone. Rather than ‘each m3 consumed’ should this be ‘m3 applied’?

Page 6 end of table 2: Water reservoir?

Page 10: no titles on x and y axis. Could the QI just be given as text instead of an inverted y-axis? Confusing graph at present.

Page 14: Figure 4A. When you have yield on the x-axis and yield as the numerator on the y-axis, is this a meaningful relationship? On page 4 it is stated that ‘All combinations were irrigated with similar annual water volumes and applying the same designed deficit irrigation strategy’ which would mean this is effectively a yield v yield graph?

Page 15: Figure 5 is a somewhat difficult to understand. Firstly, have the same symbols been used for both sets of data? At the high and low x-axis values it’s hard to tell which is which.  Given WUE seems to be more driven by yield differences than water is the relationship with WUE and QI mostly reflected a yield:quality relationship?

Secondly, as per above comment for Figure 4A , the x and y1-axis both include yield. To some extent water is in both with volume and production cost that would contribute the strong correlation.   

Page 17 first paragraph: what do you mean by ‘global’ rootstock?

Page 17: ‘Thus, grape prices can range from 50% to 125% of the base price depending on sugar content (Thornton LLP Auditor´s report Ontario Grapegrowers, 2018).’

Comment: One example from a report from Canada doesn’t really justify earlier sentence saying that ‘many wine regions..’. Just having a quick look online I also can’t readily find this reference. Section of text needs to revised with either some addition references to support this statement, or at least present them as examples.

Page 18: value of 0.53 € per kg is higher than the price for the major inland regions (SA, VIC and NSW) which have an average closer to 0.29. Tasmania is above 1.43 € per kg. Suggest using the following to cover all Australian regions.

https://www.wineaustralia.com/getmedia/807bf053-3692-448a-9ed5-c0084a47e1bb/Vintage-report-2019_full-version.pdf

Page 18: text ‘In addition, the relationships between WUEyield, production costs and QIoverall berry indicate that in current wine market conditions, maximum productive efficiency was closely related with low productive costs and QIoverall berry (Figures 4 and 5).’

Comment: is production efficiency and productive efficiency the same thing?

Round 2

Reviewer 2 Report

I'm happy to accept the manuscript in the revised form. The authors addressed the most important points and where the original text has been kept (ge. PRI v PRD) the arguements for doing so are reasonable.